# In Vitro and In Silico Studies of Neolignans from *Magnolia grandiflora* L. Seeds against Human Cannabinoids and Opioid Receptors

**DOI:** 10.3390/molecules28031253

**Published:** 2023-01-27

**Authors:** Pankaj Pandey, Mallika Kumarihamy, Krishna Chaturvedi, Mohamed A. M. Ibrahim, Janet A. Lambert, Murrell Godfrey, Robert J. Doerksen, Ilias Muhammad

**Affiliations:** 1National Center for Natural Products Research, Research Institute of Pharmaceutical Sciences, School of Pharmacy, The University of Mississippi, University, MS 38677, USA; 2Department of Chemistry and Biochemistry, The University of Mississippi, University, MS 38677, USA; 3Department of BioMolecular Sciences, Division of Medicinal Chemistry and Research Institute of Pharmaceutical Sciences, School of Pharmacy, The University of Mississippi, University, MS 38677, USA

**Keywords:** *Magnolia grandiflora*, 4-*O*-methylhonokiol, magnolol, honokiol, tetrahydromagnolol, cannabinoid, opioid, molecular docking

## Abstract

*Magnolia grandiflora* L. (Magnoliaceae) is a plant of considerable medicinal significance; its flowers and seeds have been used in various traditional remedies. Radioligand binding assays of *n*-hexane seeds extract showed displacement of radioligand for cannabinoid (CB1 and CB2) and opioid δ (delta), κ (kappa), and µ (mu) receptors. Bioactivity-guided fractionation afforded 4-*O*-methylhonokiol (**1**), magnolol (**2**), and honokiol (**3**), which showed higher binding to cannabinoid rather than opioid receptors in radioligand binding assays. Compounds **1**–**3**, together with the dihydro analog of **2** (**4**), displayed selective affinity towards CB2R (K*_i_* values of 0.29, 1.4, 1.94, and 0.99 μM, respectively), compared to CB1R (K*_i_* 3.85, 17.82, 14.55, and 19.08 μM, respectively). An equal mixture of **2** and **3** (1:1 ratio) showed additive displacement activity towards the tested receptors compared to either **2** or **3** alone, which in turn provides an explanation for the strong displacement activity of the *n*-hexane extract. Due to the unavailability of an NMR or X-ray crystal structure of bound neolignans with the CB1 and CB2 receptors, a docking study was performed to predict ligand–protein interactions at a molecular level and to delineate structure-activity relationships (SAR) of the neolignan analogs with the CB1 and CB2 receptors. The putative binding modes of neolignans **1–3** and previously reported related analogs (**4**, **4a**, **5**, **5a**, **6**, **6a**, and **6b**) into the active site of the CB1 and CB2 receptors were assessed for the first time via molecular docking and binding free-energy (∆G) calculations. The docking and ∆G results revealed the importance of a hydroxyl moiety in the molecules that forms strong H-bonding with Ser383 and Ser285 within CB1R and CB2R, respectively. The impact of a shift from a hydroxyl to the methoxy group on experimental binding affinity to CB1R versus CB2R was explained through ∆G data and the orientation of the alkyl chain within the CB1R. This comprehensive SAR, influenced by the computational study and the observed in vitro displacement binding affinities, has indicated the potential of magnolia neolignans for developing new CB agonists for potential use as analgesics, anti-inflammatory agents, or anxiolytics.

## 1. Introduction

The cannabinoid (CB) and opioid receptors are seven-transmembrane domain G-protein coupled receptors (GPCRs) known to modulate various cellular, neuronal, and cardiovascular functions [1]. Cannabinoid receptor type 1 (CB1R) antagonists/inverse agonists have the potential for treating obesity [2], obesity-related cardiometabolic disorders [3], and drug/substance abuse [4]; however, there is no such drug currently available on the market. CB1R is abundant in the central nervous system (CNS) and in peripheral tissues [5,6]. The cannabinoid receptor type 2 (CB2R) is a target for the discovery of several categories of therapeutics, such as for neuro-inflammation, cardiometabolic disorder, renal ischemia-reperfusion injury, and other diseases/disorders, including multiple sclerosis and arthritis [7,8,9,10]. Opioid receptors include three subtypes, δ- (delta), κ- (kappa), and µ- (mu), and agonism of these receptors regulates pain inhibitory pathways of the CNS [11]. Research studies have shown that both CB and opioid receptors share several pharmacological properties and act synergistically in analgesic effects at lower doses with fewer side effects [12]. Medicinal plants and their constituents have been used extensively as supplements to treat various neurological disorders, including pain, anxiety, convulsions, epilepsy, hysteria, and inflammation [13].

In the United States, *Magnolia grandiflora* L. (Magnoliaceae), commonly known as Southern magnolia, is a plant of medicinal significance that is often grown as an ornamental tree [14]. Its flowers and seeds are used in various traditional herbal remedies, including fever, rheumatism, and inflammation [15,16]. From the same genus, the bark of *M. officinalis* plays an important role in traditional Chinese and Japanese herbal medicine for the treatment of anxiety, sleep-related disorders, and allergic diseases [16]. In addition, magnolia-based products have been used for smoking cessation therapy and as aphrodisiacs, anti-depressants, and sedatives due to their hypothesized cannabimimetic and GABA-ergic-like effects [17]. Extensive research has been carried out on magnolia extracts and their major constituents to study their anti-cancer [18], anti-anxiety [19], anti-depressant [16,20], cardiovascular [21], and anti-inflammatory [22,23] activities. These pharmacological effects were proposed [16] to be primarily mediated by the presence of the neolignans 4-*O*-methylhonokiol (**1**), magnolol (**2**), honokiol (**3**), and 2′s hydrogenated derivative, tetrahydromagnolol (**4**) (Figure 1). The presence of these neolignans in the other species of the genus *Magnolia*, including *M. officinalis*, *M. obovata*, and *M. virginiana*, has led to the use of supplements prepared from these extracts as anti-depressants, analgesics, aphrodisiacs, and appetite suppressants or stimulants [17]. Earlier studies [24] have indicated that **1** and **2** could potentiate γ-aminobutyric acid A (GABA_A_) receptors and could act as positive allosteric modulators (PAMs); however, these neolignans have shown no direct interaction with the norepinephrine transporter [25]. In 2004, Kong et al., evaluated the MAO inhibitory potential of **2** and **3** using rat brain mitochondria; however, no significant rat brain MAO inhibitory activity was noted [26]. Remarkably, *M. officinalis* extract, its constituents, and their synthetic analogs have been reported as CB2 agonists and inverse agonists [27,28,29,30] and similarly as CB-related orphan receptor GPR55 inhibitors [27]; however, the safety and toxicological properties of **2**, **3**, and bark extracts of *M. officinalis* and *M. obovate* have been reported as safe for consumption [31].

In a continuation to our research efforts to establish potential leads towards human cannabinoids and opioid receptors, *M. grandiflora* seed extract showed significant binding affinities to CB and opioid receptors, and therefore a detailed investigation of its active constituents and their structure-activity relationships (SAR) has been conducted using in silico studies that utilize the CB2 X-ray crystal structure. A bioassay-guided isolation of the hexane extract, which showed significant displacement of radioligand in cannabinoid and opioid receptors (64% CB1, 74% CB2, δ 93%, κ 61%, and µ 85%), yielded 4-*O*-methylhonokiol (**1**), magnolol (**2**), and honokiol (**3)** with selective CB2 activity (K*_i_* 0.29, 1.40, and 1.94 μM, respectively). The structures of these neolignans are closely related to the structures of the potent CB1 agonist Δ^9^-tetrahydrocannabinol (Δ^9^-THC) and the CB1/CB2 non-selective agonist CP55,940 (Figure 1). In addition to CB and opioid activity, we report the first systematic in silico molecular docking studies of neolignans (**1**–**4**) and their structurally-related previously reported analogs (**4a**, **5**, **5a**, **6**, **6a**, **6b**) [29], using the active-state X-ray crystal structures of CB1R and CB2R to gain an understanding of the various protein–ligand interaction patterns, their putative binding modes, and the observed SAR of these ligands. A detailed understanding of the SAR of the isolated metabolites, as well as the additive effect between compounds **2** and **3,** has been reported for the first time, as supported via the in vitro data and the in silico docking studies that take advantage of the first CB2 X-ray crystal structure, which was released in 2020 [32].

## 2. Results and Discussion

The *n*-hexane extract of *M. grandiflora* showed significant radioligand displacement of CB1, CB2, δ, κ, and µ opioid receptors (Table 1). The extract was subjected to centrifugal preparative thin-layer chromatography (CPTLC; for details, see Experimental Section) to afford 4-O-methylhonokiol (**1**), magnolol (**2**), and honokiol (**3**), together with marked fatty acids. The structures of the isolated compounds were identified based on 1D- and 2D- nuclear magnetic resonance (NMR) and electrospray ionization mass spectrometry (ESI-MS). Compounds **1**–**3**, as well as **4** (tetrahydromagnolol, purchased from Sigma-Aldrich), were evaluated using in vitro binding assays with CB and opioid receptors (Table 1 and Table 2 and Figure 1, Figure 2 and Figure 3). The preliminary results determined at a concentration of 10 µM revealed strong displacement by **1** towards both CB1R and CB2R (i.e., 91.3% and 82.2%**,** respectively). The mixture of **2** + **3** (1:1) showed similar displacement of radioligand (99.8% and 91.0%, respectively), compared to either compound **2** or **3** alone (Table 1). Based on the secondary assay results, compounds **1***–***3** displayed strong selective affinity towards CB2R compared to CB1R, with K*_i_* values of **1** towards CB2R as being 0.29 ± 0.022 μM, which is significantly better than those of **2** and **3** (K*_i_* 1.44 ± 0.138 and 1.94 ± 0.162 µM, respectively), while tetrahydromagnolol (**4**) showed similar displacement to that of **2** (Figure 2 and Figure 3).

The hexane extract displayed significant radioligand displacement at all three opioid receptors: δ (93%), κ (60.8%), and µ (84.5%); however, all the isolated neolignans (**1**–**3**) were devoid of binding affinity at δ and κ. Nevertheless, the isolated neolignans showed high micromolar binding affinity at the µ receptor. Indeed, a 1:1 mixture of **2** and **3** showed better additive percentage displacement (%) of radioligand at δ (93.3%), κ (59.2%), and µ (74.0%) receptors, compared to **1**–**3** alone, but was similar to those of the hexane extract (Table 1), which provides an explanation for the strong displacement activity of the n-hexane extract. Additionally, **2** and **3** showed a low affinity for µ opioid receptors during IC_50_ and K*_i_* determinations (Figure 4).

CB binding affinities of compounds **1**–**4** have been previously reported [27,28,29], and are in accordance with the results obtained for CB2R in our laboratory. Furthermore, compounds **1**–**4** were reported to possess binding affinities (K*_i_*) between 2*–*9 μM for CB1R [27,28,29]. However, in our lab, this observation was only true for 4-O-methylhonokiol (**1**); the remaining three compounds possessed binding affinities (K*_i_*) between 14*–*20 μM (Table 2). Rempel et al. (2013) have reported magnolol (**2**), and tetrahydromagnolol (**4**) as partial CB2R agonists (K*_i_* 1.44 and 0.41 μM, respectively), and honokiol (**3**) was considered as a CB2R antagonist or inverse agonist (K*_i_* 5.61 μM) [27]. Schuehly et al. (2011) reported intriguing nonspecific heteroactive behavior of 4-O-methylhonokiol (**1**) at the CB2R as an inverse agonist at G_i/o_ and as a full agonist regarding intracellular Ca^2+^ [Ca^2+^]_i_. In contrast, Fuchs et al. (2013) reported **1** as an agonist at both CB receptor subtypes in forskolin-induced cAMP (3*’*,5*’*-cyclic adenosine monophosphate) accumulation assays [29]. Our CB binding affinity data of **1*–*4** are in close agreement with those reported previously [27,28,29].

### Protein-Ligand Interaction Study

Due to the unavailability of an NMR or X-ray crystal structure of bound neolignans with the CB1 and CB2 receptors, a docking study was performed to predict ligand–protein interactions at a molecular level. The in silico docking studies were performed by taking advantage of the first CB2 X-ray crystal structure (PDB ID: 6KPC) [32] which was released in 2020 and the CB1 receptor (PDB ID: 5XRA) [33]. The isolated neolignans (**1**–**4**) and structurally-related previously published compounds (**4a**, **5**, **5a**, **6**, **6a**, and **6b**) [29] were studied to understand the SAR using docking and binding free-energy calculations. The docking protocol was validated by a self-docking approach in which native ligands, 8D3 and E3R, were docked into their corresponding protein structures, CB1 and CB2, respectively. Further, we calculated the Root Mean Square Deviation (RMSD) between docked poses and experimental poses of native ligands with the CB1 and CB2 receptors. The overlay of experimental poses of native ligands with the docked poses showed an identical conformation, with very small RMSD differences of 0.35 Å and 0.64 Å, respectively. The binding affinity data from Table 3 indicates that the replacement of one hydroxyl by a methoxy group (**4a**) in tetrahydromagnolol (**4**) led to a significant increase in CB1R affinity. Simultaneously, the selectivity towards CB2R was lost. Our computational results also justified this SAR through analysis of the binding free energy data. The replacement of one hydroxyl by a methoxy group in **4a** showed better-predicted binding free-energy (ΔG = −76.78 kcal/mol) compared to tetrahydromagnolol (**4**) (ΔG = −71.06 kcal/mol) at the CB1R.

The replacement of the hydroxyl group with the methoxy group resulted in a steric clash of the methoxy moiety with Phe170. Thus, the methoxy-containing phenyl ring of **4a** is inverted towards Phe200, and the alkyl chain (*n* = 3) is oriented towards the small hydrophobic pocket in a similar fashion to the cocrystallized ligand AM1152 in the 5XRA crystal structure wherein the dimethyl heptyl moiety is positioned in that pocket (Figure 5). The strong hydrophobic interaction of the methoxy group of **4a** with Phe200 and the other lesser hydrophobic interactions of the propyl chain with Leu193, Val196, Phe200, Leu276, Trp279, Met363, Cys386, and Leu387 support a better CB1R affinity compared to **4**.

Like methoxytetrahydromagnolol (**4a**), the methylated compound **5a**, which differs at the R2 position (pentyl chain), displayed a 21-fold higher binding affinity towards CB1R than the parent biphenyl **5**, which is clearly explained by binding free-energy data. Similarly, compound **5a** exhibited better binding free-energy (ΔG = −83.77 kcal/mol) compared to **5** (ΔG = −78.78 kcal/mol).

The docking orientation of **5** and **5a** is very similar in the active site of the CB1R (Figure 6), except for the pentyl and propyl chains, which are interchanged. Like compound **4a**, the strong hydrophobic interactions between Phe200 and Cys386 of CB1R and the methoxy moiety of **5a** were observed, which might explain the better affinity of **5a** to CB1R compared to **5**.

The impact of the methoxy group on CB1R affinity was found to be more pronounced compared to the CB2R. When the methoxy group was introduced in the para-position of the short propyl residue of **6,** a remarkable enhancement in CB1R affinity (a 15-fold increase) was observed (**6a**).

The docking poses of **6** and **6a** in the active site of CB1R are markedly similar (Figure 7), and the alkyl chain is also oriented in the same manner, which was different in the docking pose of **5** and **5a**. The additional strong hydrophobic interactions between the methoxy moiety of **6a** and Phe200, Phe170, and Cys386 (also H-bonding) of CB1R might explain the increased affinity (K*_i_* = 0.00957 µM) of **6a** to CB1R compared to **6**. Compared to the unmethylated **5** and **6,** the methylated **5a** and **6a** exhibited strong predicted CB1R affinity (Figure 6 and Figure 7C), while predicted CB2R binding affinity was almost unchanged (Figure 6D and Figure 7D); however, the introduction of a methoxy group in the para-position to the hexyl residue (**6b**) drastically reduced the affinity for the CB2R (from K*_i_* = 0.0294 µM to 0.234 µM, Figure 7C). Interestingly, **6a**, with a functional group rearrangement (methoxy group in the para-position with respect to the propyl residue), displayed a 12-fold increase in CB2R affinity (K*_i_*= 0.0238 µM). Analysis of the docking pose of **6a** and **6b** in the active site of CB2R clearly explained the activity difference between **6a** and **6b**. In **6a**, the presence of the methoxy group oriented towards Thr114 (distance Thr (O–H) and –O–CH_3_ = 4.27 Å), the unmethylated hydroxyl group forming H-bonding with Ser285, and the biphenyl contributing to π–π stacking with Phe87 and Phe183 allowed the alkyl chain (*n* = 6) to be directed towards the small hydrophobic pocket similar to the cocrystallized ligand AM1152 in the 5XRA crystal structure that positions the dimethyl heptyl moiety in that pocket. This also afforded lower binding free-energy (ΔG = −80.56 kcal/mol) compared to **6b** (ΔG = −69.06 kcal/mol); however, in **6b**, the methoxy moiety moved slightly upwards towards Ser90, and the alkyl chain (*n* = 3) was redirected towards the small hydrophobic pocket similar to the cocrystallized ligand AM1152 in the 5XRA crystal structure [33]. The identical orientation of the hexyl alkyl chain of **6a** towards the hydrophobic pocket where the alkyl chain of the cocrystallized ligand AM1152 in the 5XRA CB1 crystal structure is situated led to the strong activity of **6a** compared to **6b**.

Both honokiol (**3**) and 4-*O*-methylhonokiol (**1**) docked in similar orientations within the active site of CB1R and showed strong π–π interactions with Phe170/Phe174 and Phe268, as well as H-bonding with one of the hydroxyls of **3** and **1** with Ser383 (Figure 8). Furthermore, the methoxy group of **1** exhibited additional strong hydrophobic interactions with Ser173 and Phe177, which led to better binding free-energy (ΔG = −78.20 kcal/mol) compared to **3** (ΔG = −71.19 kcal/mol). These observed data do not match with Fuchs et al. (2013) [29] experimental in vitro data, whereas **3** (K*_i_* = 6.46 µM) is more active at CB1R than **1** (K*_i_* = 8.34 µM). Interestingly, our in-house testing of the binding affinity of **1** (K*_i_* = 3.85 µM) and **3** (K*_i_* = 14.55 µM) against CB1R matches closely with the computational binding free-energy data. In the case of CB2R, 4-*O*-methylhonokiol (**1**) and honokiol (**3**) docked in a similar fashion; however, 4-*O*-methylhonokiol (**1**) shifted ~1.6 Å intracellularly from the position of the honokiol (**3**) atoms, which allows this molecule (**1**) to form strong hydrogen bonding (through the phenolic hydroxyl moiety) with Ser285 and strong π–π interactions with Phe87 and Phe183. These interactions were absent in the honokiol (**3**) docking pose with CB2R. The only interactions observed between honokiol (**3**) and CB2R were hydrophobic. Remarkably, the binding free-energy (ΔG) data matches well with the experimental activity data for these two molecules at the CB2R.

## 3. Materials and Methods

### 3.1. Extraction and Bioassay-Guided Isolation of Compounds

Mature seeds of *M. grandiflora* were collected at the University of Mississippi (MS 38,677) in November 2013. The voucher specimen (NCNPR #15,895) is deposited at the University of Mississippi. The air-dried seeds (107 g) were powdered and extracted with *n*-hexane (200 mL × 2 for 24 h) followed by 95% EtOH. The combined hexane extracts were evaporated under reduced pressure. Two grams of hexane extract were chromatographed over a centrifugal preparative thin layer chromatographer (CPTLC, Chromatotron^®^, Analtech Inc., Newark, DE, USA) using a 6 mm silica gel rotor. The sample was dissolved in dichloromethane (DCM), applied to the rotor, and then eluted with *n*-hexane, followed by DCM and MeOH (200 mL each) to yield eighteen fractions. These fractions later yielded three major lignans, 4-*O*-methylhonokiol (**1**, 36 mg), honokiol (**2**, 20 mg), and magnolol (**3**, 15 mg), together with marked fatty acids. All fractions were monitored and collected via TLC analysis (silica gel; solvents: *n*-hexane-EtOAc; 75:25).

4-*O*-methylhonokiol (**1**); UPHPLC/APCI-MS *m*/*z* 281.3 ([M + H])^+^ C_19_H_20_O_2_ + H; the ^1^H and ^13^C NMR were indistinguishable to those reported [34].

Honokiol (**2**); UPHPLC/APCI-MS *m*/*z* 267.3 ([M + H])^+^ C_18_H_18_O_2_ + H; the ^1^H and ^13^C NMR were indistinguishable to those reported [35].

Magnolol (**3**); UPHPLC/APCI-MS *m*/*z* 267.3 ([M + H])^+^ C_18_H_18_O_2_ + H; the ^1^H and ^13^C NMR were indistinguishable to those reported [35].

### 3.2. Cannabinoid and Opioid Receptor Binding Assay

#### 3.2.1. Reagents

CP55,940 was purchased from Tocris Bioscience (Minneapolis, MN, USA). BSA, Trizma^TM^ hydrochloride (Tris-HCl), penicillin, streptomycin, and nonenzymatic cell dissociation solution were purchased from Sigma-Aldrich (St. Louis, MO, USA). Radioligands, GF/C, GF/B 96-well plates, and MicroScint^TM^-20, were purchased from PerkinElmer (Waltham, MA, USA). Membrane preparation was made using a 50 mM Tris-HCl buffer with pH 7.4. Tetrahydromagnolol (**4**) was purchased from Sigma-Aldrich (St. Louis, MO, USA) (% purity ≥ 95).

#### 3.2.2. Cell Culture and Membrane Preparation

Human embryonic kidney 293 (HEK293) cells (ATCC) were stably transfected with cannabinoid receptor subtypes 1 and 2 and maintained and harvested as described [36,37]. HEK293 cells stably transfected with δ, κ, and μ opioid subtypes were a generous gift from Roth Laboratories (University of North Carolina at Chapel Hill, NC, USA). Opioid cells were maintained as previously described [37,38]. Membranes were made by washing the cells with cold PBS. The cells were then scraped in cold 50 mM Tris-HCl, pH 7.4 buffer. The solution was centrifuged at 5200× *g* for 10 min at 4 °C. Next, the supernatant was discarded, and the pellet was washed with more Tris-HCl buffer, homogenized via Sonic Dismembrator (Fisher Scientific, Pittsburgh, PA, USA) and then centrifuged at 24,000× *g* for 40 min at 4 °C. Finally, the pellet was re-suspended in cold 50 mM Tris-HCl buffer, aliquoted into 2 mL vials, and stored at −80 °C. The total membrane protein concentration was measured using a Pierce BCA Protein Assay Kit (Thermo Scientific, Rockford, IL, USA) as per the manufacturer’s protocol.

#### 3.2.3. Competitive Radioligand Binding Assays

Cannabinoid and opioid competitive radioligand binding assays were performed as previously described [5,36,37,38]. Saturation experiments were performed for all the receptors to determine receptor concentration and radioligand dissociation constant (K_d_) for the membrane. Percent displacements were evaluated for all the cannabinoid and opioid subtypes with a triplicate of a fixed concentration (10 µg/mL for extracts and fractions, 10 μM for purified compounds). The samples competed with a tritium-labeled ligand with a known affinity of the receptor of interest-{[^3^H]-CP55,940 for CB1R and CB2R, [^3^H]-U-69,593 for *κ*, [^3^H]-DAMGO for *μ*, or [^3^H]-enkephalin (DPDPE) for *δ*}, with the radioligand concentration equal to its K_d_. Control/test compounds were dissolved in DMSO at 10 μg/mL for extracts and fractions and 10 μM for purified compounds. Dilutions of the membrane, radioligand, and control/test compounds were made in a Tris-EDTA buffer (50 mM Tris-HCl (pH 7.4), 20 mM EDTA, 154 mM NaCl, and 0.2% fatty-acid free BSA), with pH = 7.4 for cannabinoids and 50 mM Tris-HCl (pH 7.4) for opioids. The competitive binding assays were performed using 12 serial dilutions of each compound ranging from 0.002–300 μM (control compounds were serially diluted from 10 μM to 0.06 nM). The cannabinoid assays were incubated for 90 min at 37 °C with gentle agitation. The opioid assays were incubated for 60 min at room temperature. Bound radioligand was collected on GF/C (cannabinoid) or GF/B plates (opioid), washed 10 times with ice-cold 50 mM Tris-HCl (pH 7.4)/0.1% BSA (cannabinoid) or ice-cold 50 mM Tris-HCl (pH 7.4) (opioid). Radiodetection was measured with 50 µL (cannabinoid) or 25 µL (opioid) MicroScint^TM^-20 on a TopCount NXT HTS Microplate Scintillation Counter (PerkinElmer, Waltham, MA, USA). The IC_50_ and K*_i_* values were calculated by a non-linear curve fit model using GraphPad Prizm 5.0 software (GraphPad Software Inc., San Diego, CA, USA). Each compound was tested in triplicate unless stated otherwise.

Percent displacement [5] was calculated to represent the ability of the samples to displace the radioligand binding for a given cannabinoid or opioid receptor subtype.

% displacement was calculated as follows:$$100-\frac{\;\mathrm{binding}\;\mathrm{of}\;\mathrm{compound}-\mathrm{nonspecific}\;\mathrm{binding}}{\mathrm{specific}\;\mathrm{binding}}\times 100$$

### 3.3. Computational Method

The X-ray crystal structures of the active-state of cannabinoid receptors 1 (PDB ID: 5XRA) [33] and 2 (PDB ID: 6KPC) [32] were downloaded from the Protein Data Bank website. CB1 and CB2 protein structures were prepared by adding hydrogen atoms, bond orders, and missing side chain residues and by proper ionization at a physiological pH of 7.4 using the Protein Preparation Wizard (PPW) [39] module implemented in the Schroödinger software. We used CP55,940 as a reference compound for the current study. The ligands magnolol **(2**), CP55,940, honokiol (**3**), tetrahydromagnolol (**4**), 4-*O*-methylhonokiol (**1**), **4a**, **5**, **5a**, **6**, **6a**, and **6b** were sketched in Maestro [Schrödinger Release 2020-4: Maestro, Schrödinger, LLC, New York, NY, USA, 2020] and energy-minimized with the LigPrep [Schrödinger Release 2020-4: LigPrep, Schrödinger, LLC, New York, NY, USA, 2020] module of the Schrödinger suite using the OPLS3e (optimized potential for liquid simulations 3e) force field [40]. The grids for CB1R and CB2R were prepared using the centroid of the co-crystalized ligands in the respective X-ray structures of CB1R and CB2R. The van der Waals radius-scaling factor and partial charge cutoff were maintained at 1 and 0.25, respectively. No additional constraints were used when preparing the grid or for docking. The docking of the ligands into the active states of CB1R and CB2R was performed using the Extra Precision (XP) [41] method of Glide [42] using the OPLS3e force field [40]. The docking protocol was validated by redocking of native co-crystallized ligands of CB1 and CB2 receptors in their corresponding protein structures. During docking, ligand sampling was kept flexible while the receptor (protein) was kept rigid. After docking, the binding free energies (Prime MM-GBSA free energies) [43] of the docked structures were calculated using the Prime [43] module of the Schrödinger software.

## 4. Conclusions

Bioassay-guided isolation revealed that neolignans **1**–**3** from *M. grandiflora* seeds and an analog of **2** (**4**) displayed various degrees of displacement affinities against cannabinoid and opioid receptors. These neolignans were previously reported as CB2 agonists [29]. The observed in vitro displacement binding affinities and previously published functional data provided further evidence that neolignan **1** acts as a promising CB2 agonist. An in-depth understanding of the SAR of the isolated compounds as well as the additive effect between compounds **2** and **3** has been reported for the first time, supported via the in vitro data and the in silico docking studies that take advantage of the first CB2 X-ray crystal structure, which was released in 2020. Methylation of **2** at the *p*-position and the replacement of the hydroxyl group with a methoxy substituent increase the displacement activity towards CB1 as shown in Table 1. Similarly, an equal (1:1) mixture of **2** and **3** showed additive displacement activity towards the tested receptors as compared to either **2** or **3** alone, which in turn provides an explanation for the strong displacement activity of the *n*-hexane extract. The in silico docking studies of these neolignans (**1**–**4**) and their related previously reported analogs (**4a**, **5**, **5a**, **6**, **6a**, and **6b**) [29] with the active states X-ray crystal structures of CB1R and CB2R revealed the putative binding mechanism, selectivity, and interaction profile of the protein–ligand complex. The docking and binding free-energy results indicate that one of the hydroxyl moieties of the molecules in the present study formed strong H-bonding through Ser383 and Ser285 with CB1R and CB2R, respectively. The impact of the methoxy group on the affinity towards CB1R and CB2R was explained in terms of binding free-energy data and the orientation of the alkyl chain within the CB1R. The additional strong hydrophobic interactions between the methoxy moiety of **5a** and **6a**, and Phe200 and Cys386 of CB1R, as well as the orientation of the pentyl and propyl chains, are interchanged across **5** and **5a**, which could explain the increased affinity towards CB1R compared to the CB2R. We believe that insights gained from this study could provide a platform for medicinal chemists working in this important area of cannabinoid research to utilize a new scaffold for the design of new analogs from neolignans. This, in turn, can lead to the identification of new synthetic compounds with improved affinity, functional activity, and/or selectivity for the CB receptors.

Our findings suggest the potential utility of Magnolia neolignans and their derivatives for the development of new CB agonists for use as analgesic or anti-inflammatory lead compounds. Furthermore, the combined effect of potent cannabinoid (CB) and weak μ opioid binding affinity of a mixture of magnolol and honokiol analogs offers a new window for the treatment of opioid dependence and opioid withdrawal symptoms; however, further in vivo studies should be implemented to help deepen the understanding of the mechanisms of action of these new lead natural compounds.

## Figures and Tables

**Figure 1 molecules-28-01253-f001:**
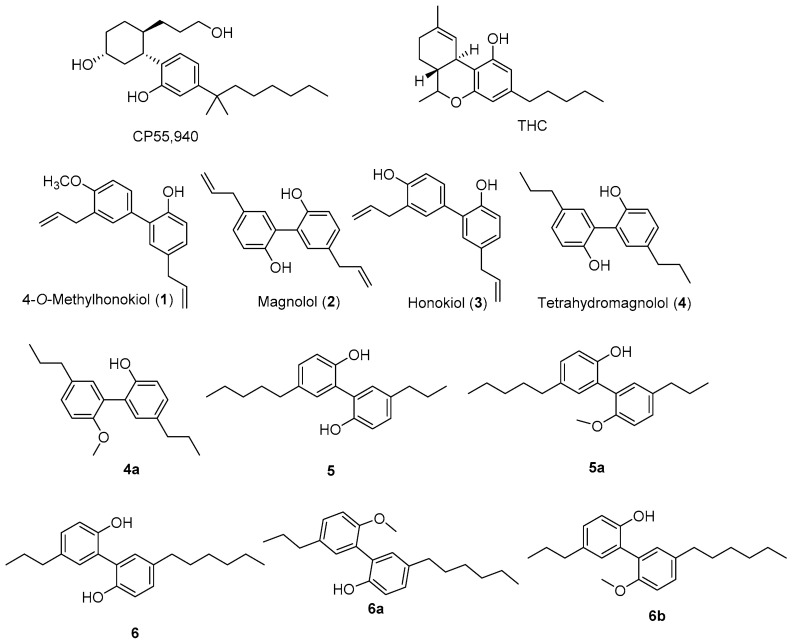
Compounds isolated/identified (**1**–**4**) from *M. grandiflora* seed and related (**4a**–**6b**) compounds [29] were used for molecular docking studies.

**Figure 2 molecules-28-01253-f002:**
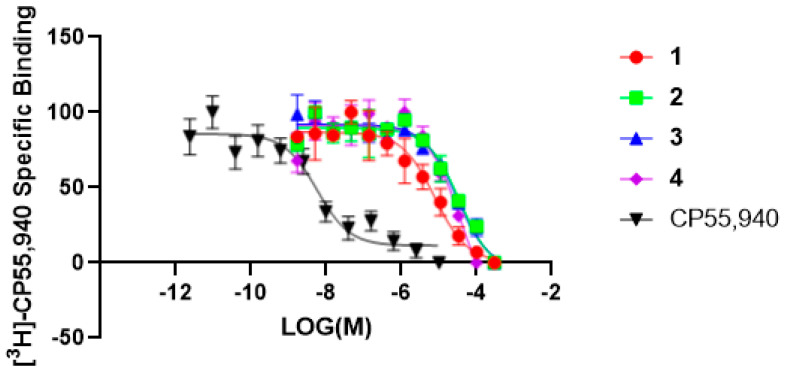
The radioligand displacement curves were obtained for compounds (**1**–**4**) for the CB1R. CP55,940 was used as a positive control.

**Figure 3 molecules-28-01253-f003:**
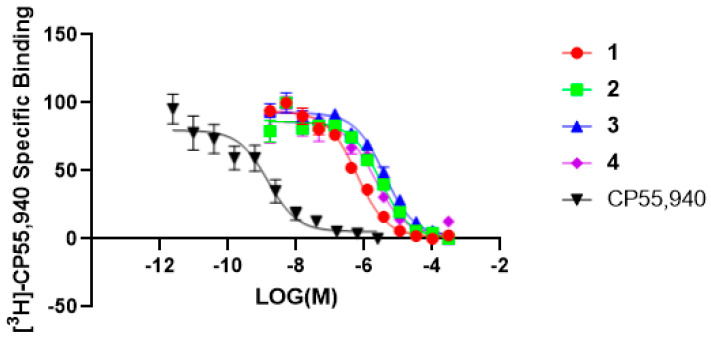
The radioligand displacement curves were obtained for compounds (**1**–**4**) for the CB2R. CP55,940 was used as a positive control.

**Figure 4 molecules-28-01253-f004:**
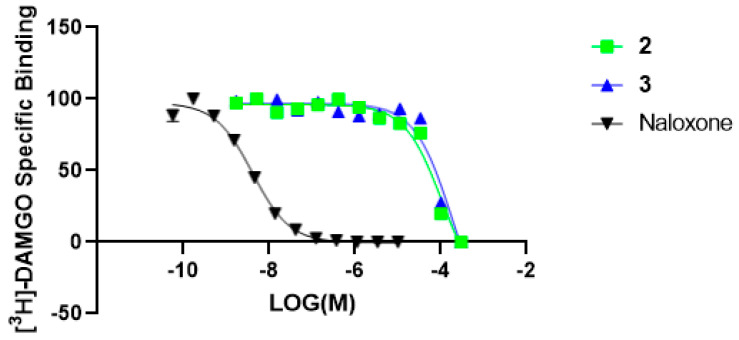
The radioligand displacement curves were obtained for compounds (**2** and **3**) for the µ opioid receptor. Naloxone hydrochloride was used as a positive control.

**Figure 5 molecules-28-01253-f005:**
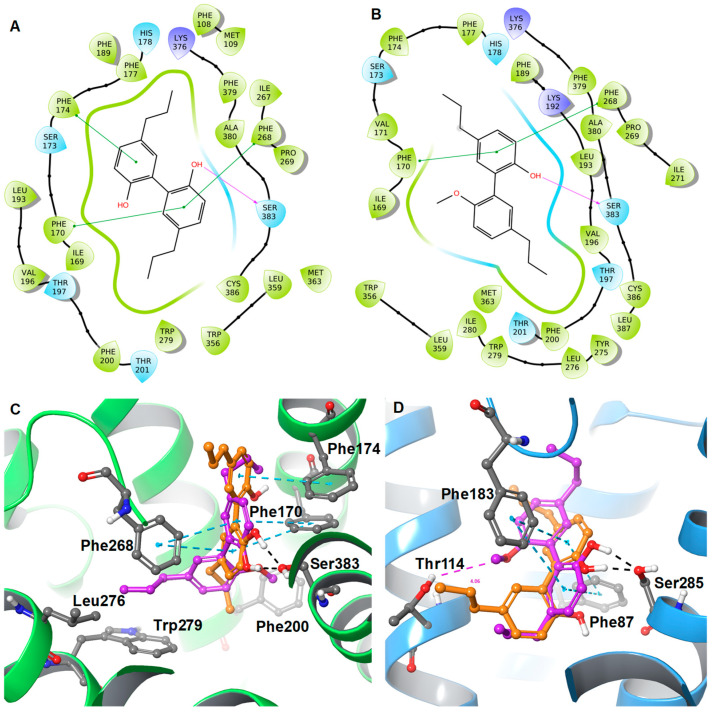
2D interaction diagrams of (**A**) **4** and (**B**) **4a** along with the 3D overlaid representations of (**C**) **4** (carbon in orange) with **4a** (carbon in plum) against the CB1R and (**D**) **4** (carbon in orange) with **4a** (carbon in plum) against the CB2R. The key residues are shown in the ball and stick model (carbon in grey).

**Figure 6 molecules-28-01253-f006:**
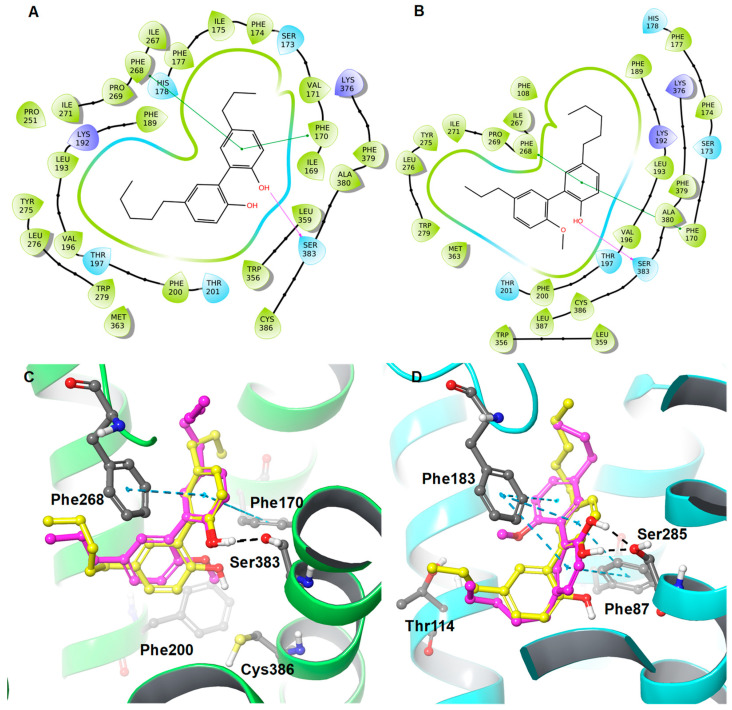
2D interaction diagrams of (**A**) **5** and (**B**) **5a** along with the 3D overlaid representations of (**C**) **5** (carbon in yellow) with **5a** (carbon in magenta) against the CB1R and (**D**) **5** (carbon in yellow) with **5a** (carbon in magenta) against the CB2R. The key residues are shown in the ball and stick model (carbon in grey).

**Figure 7 molecules-28-01253-f007:**
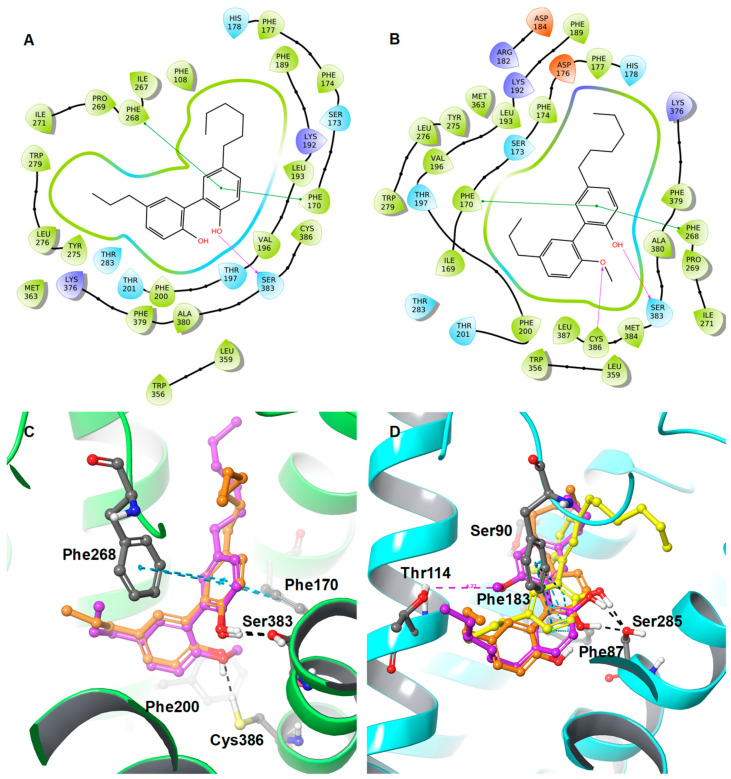
2D interaction diagrams of (**A**) **6** and (**B**) **6a**, along with the 3D overlaid representations of (**C**) **6** (carbon in orange) with 6a (carbon in plum) against the CB1R and (**D**) **6** (carbon in orange) with **6a** (carbon in plum) and 6b (carbon in yellow) against the CB2R. The key residues are shown in the ball and stick model (carbon in grey).

**Figure 8 molecules-28-01253-f008:**
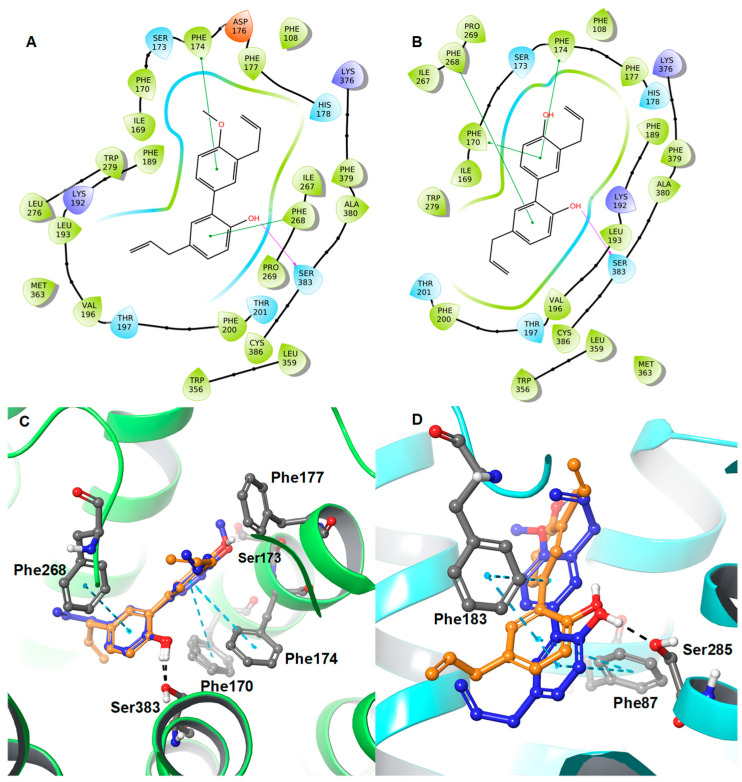
2D interaction diagrams of (**A**) **1** and (**B**) **3** along with the 3D overlaid representations of (**C**) **1** (carbon in blue) with **3** (carbon in orange) against the CB1R and (**D**) **1** (carbon in blue) with **3** (carbon in orange) against the CB2R. The key residues are shown in the ball and stick model (carbon in grey).

**Table 1 molecules-28-01253-t001:** The binding affinities of extracts/compounds towards human cannabinoids and opioid receptors.

10 μM Compounds or 10 μg/mL Extracts *	Cannabinoid Receptors (% Displacement)			Opioid Receptors (% Displacement)	
	CB1	CB2	Δ	*κ*	*µ*
*n*-Hexanes extract	63.8	73.7	93.0	60.8	84.5
Ethanol extract	55.4	94.7	76.2	23.1	72.6
4-*O*-Methylhonokiol (**1**)	91.3	82.2	31.4	18.3	46.2
Magnolol (**2**)	52.7	74.7	7.3	10.7	52.9
Honokiol (**3**)	50.8	65.7	NA	NA	54.2
**2** + **3** (1:1)	99.8	91.0	93.3	59.2	74.0
Tetrahydromagnolol (**4**)	30.8	78.2	NA	NA	NA
CP55,940	82.5	101.3	NT	NT	NT
Naloxone	NT	NT	97.0	100.2	99.8

* All purified compounds and **2** + **3** were tested at a concentration of 10 μM. For the cannabinoid binding assay, the CB agonist CP55,940 was used as the positive control. For the opioid receptor binding affinity assay, the opioid receptor antagonist naloxone was used as the positive control. NA = Not active (no displacement), NT = Not tested.

**Table 2 molecules-28-01253-t002:** The binding affinities (K*_i_* and IC_50_) of selected compounds against CB1R, CB2R, and µ opioid receptors.

Compound	CB1R (μM)		CB2R (μM)		µ Opioid Receptor (μM)	
	IC_50_	K*_i_* ± SEM	IC_50_	K*_i_* ± SEM	IC_50_	K*_i_* ± SEM
**1**	7.69	3.85 ± 0.89	0.59	0.29 ± 0.02	n/a	n/a
**2**	35.64	17.82 ± 3.43	2.89	1.40 ± 0.14	106.20 *	53.12 *
**3**	29.11	14.55 ± 2.47	3.88	1.94 ± 0.16	91.06 *	182.10 *
**4**	38.17	19.08 ± 0.79	1.99	0.99 ± 0.14	NA	NA
CP55,940 ^#^	0.006859	0.0033 ± 0.00128	0.00287	0.001439 ± 0.00027	NT	NT
Naloxone ^	NT	NT	NT	NT	0.00409	0.002049 ± 0.000179

* Run as a singlet (no S.E.M. calculated). ^#^ For the cannabinoid binding assay, the CB agonist CP55,940 was used as the positive control. ^ For the opioid receptor binding affinity assay, the opioid receptor antagonist naloxone was used as the positive control. IC_50_; the concentration required for 50% displacement of ^3^H-labeled ligand. n/a; not applicable. NA = Not active, NT = Not tested.

**Table 3 molecules-28-01253-t003:** Summary of docking scores and binding free energies (ΔG) of Magnolia compounds to the CB1 and CB2 receptors.

Compound	CB1R Agonist K*_i_* (nM) *	CB2R Agonist K*_i_* (nM) *	CB1R		CB2R	
			GlideScore (kcal/mol)	ΔG (kcal/mol)	GlideScore (kcal/mol)	ΔG (kcal/mol)
CP55,940	1.28	1.42	−12.045	−85.71	−12.156	−81.21
**2** (Magnolol)	3150	1440	−10.243	−74.55	−9.765	−64.48
**4** (Tetrahydromagnolol)	2260	416	−10.346	−71.06	−11.194	−73.41
**3** (Honokiol)	6460	5610	−10.854	−71.17	−8.989	−59.73
**1** (4-*O*-methylhonokiol)	8340 ± 3200	43.3 ± 17.1	−11.106	−78.20	−10.564	−73.06
**4a**	267 ± 58	221 ± 57	−11.275	−76.78	−10.251	−71.46
**5**	362 ± 113	37.5 ± 7.8	−12.307	−78.78	−11.195	−77.88
**5a**	17.3 ± 1.4	31.0 ± 9.9	−11.652	−83.77	−10.401	−76.00
**6**	145 ± 48	29.4 ± 9.0	−12.228	−80.02	−11.577	−81.06
**6a**	9.57 ± 5.43	23.8 ± 7.1	−12.311	−88.39	−11.224	−80.56
**6b**	313	281 ± 101	ND	ND	−10.944	−69.06

* Data used here are obtained from the original publication [29]. ND = not determined.

## Data Availability

Not applicable.

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
