# Peer review of "In Vitro and In Silico Studies of Neolignans from Magnolia grandiflora L. Seeds against Human Cannabinoids and Opioid Receptors"

_molecules, 2023, doi:10.3390/molecules28031253_

Round 1

Reviewer 1 Report

the MS have similarity index of 33% which is higher than normal, should be less than 19%

Numbers of references are missing, should be included. avoid citing old references,

Last paragraph of the introduction should explain the need of the study, the questions which are targeted to answer, you novelty, what has been done in the past should be mentioned and what new you have been reported. otherwise the study is not effective if the similar work has been published or done.

Material and method should have enough citations updated one or original one.

result need to be improve , poorly represent the data.

discussion is not strong, keep citing of top class articles help you to strengthen you discussion section. 

Conclusion is not complete, what you have done, what question need to be answer. 

people should have home take message. the overall structure need to be rearranged. 

Author Response

RESPONSE TO REVIEWERS COMMENTS AND SUGGESTIONS – REVIEWER # 1

(All revisions were carried out using track change)

  1. The MS have similarity index of 33% which is higher than normal, should be less than 19%.

Response: Thank you very much for your valuable suggestion. We checked the manuscript carefully and rewrote some sentences to avoid similarities in the introduction and methods sections. Also, we updated the citations for the introduction and methods sections to make sure they are complete and current. The entire manuscript has now been checked carefully and edited accordingly to avoid any grammar mistakes and English language issues.

  1. Numbers of references are missing, should be included. avoid citing old references.

Response: We have added citations where relevant and included some more recent references in the revised manuscript.

  1. Last paragraph of the introduction should explain the need of the study, the questions which are targeted to answer, you novelty, what has been done in the past should be mentioned and what new you have been reported. otherwise the study is not effective if the similar work has been published or done.

Response: We have updated the last paragraph of the Introduction in the revised manuscript as suggested. The main novelty and future direction of this study have been highlighted in the revised manuscript as follows: "A detailed understanding of the SAR of the isolated compounds as well as the additive effect between compounds 2 and 3 has been reported for the first time, as supported via the in vitro data and the in silico docking studies that take advantage of the first CB2 X-ray crystal structure, which was released in 2020”.

We also included the following sentence in the Conclusion section to make a more clear and concise summary of our current work. “Methylation of 2 at the p-position and the replacement of the hydroxyl group with a methoxy substituent increase the displacement activity towards CB1 as shown in Table 1. Similarly an equal (1:1) mixture of 2 and 3 showed additive displacement activity towards the tested receptors compared to either 2 or 3 alone, which in turn provides an explanation for the strong displacement activity of the n-hexane extract”.

  1. Material and method should have enough citations updated one or original one.

Response: We have added more citations for the Materials and Methods section in the revised manuscript.

  1. result need to be improve , poorly represent the data.

Response: The Results section has been revised as suggested by the reviewer.

  1. discussion is not strong, keep citing of top class articles help you to strengthen you Discussion section. 

Response: We have updated the Discussion section in the revised manuscript as suggested.

  1. Conclusion is not complete, what you have done, what question need to be answer. 

Response: The Conclusion section has been adjusted to cover the novelty of our work and the potential future direction. Please refer response to query 3, above.

  1. people should have home take message. the overall structure need to be rearranged. 

Response: The take-home message has been incorporated in the revised Conclusion section. Also, we have revised the manuscript carefully to meet the reviewer’s standards. We believe that the constructive comments of the reviewer significantly improved the quality of our manuscript.

Reviewer 2 Report

The manuscript entitled "In vitro and In silico studies of neolignans from Magnolia grandiflora L. seeds against human cannabinoids and opioid receptors " by Pandey et al is well written, well presented and in my views it should be accepted for publication in Molecules

Author Response

RESPONSE TO REVIEWERS COMMENTS AND SUGGESTIONS – REVIEWER # 2

(All revisions were carried out using track change)

The manuscript entitled "In vitro and In silico studies of neolignans from Magnolia grandiflora L. seeds against human cannabinoids and opioid receptors " by Pandey et al is well written, well presented and in my views it should be accepted for publication in Molecules.

Response: Thank you. We appreciate very much the positive feedback of the reviewer that reflects the interest of our study to many readers.

Reviewer 3 Report

The manuscript reports the investigation of the cannabinoid affinity of the neolignans extracts from Magnolia grandiflora. While appreciating the technical soundness of the study and the clarity of the results, the reviewer has some doubts regarding the novelty of the reported results since the presence of these neolignans into the extracts from Magnolia species was already reported and their activity on the cannabinoid receptors was already document along with the study on the affinity of these biphenol derivatives on other therapeutically relevant targets such as GABAA and GPR55 receptors, kinases and other enzymes. Probably the only original part is the molecular docking but the reviewer is not sure that it can render the paper publishable. 

Concerning the modelling studies, they are technically correct and the results are described in detail. The reviewer would suggest to start the docking analyses with a redocking study of the bound ligands to assess if the utilized computational approach is able to reproduce the experimental structure. Finally, the reviewer thinks that the manuscript can be considered only if the Authors convincingly explain which is the novelty of their study compared to the already published data.

Author Response

RESPONSE TO REVIEWERS COMMENTS AND SUGGESTIONS – REVIEWER # 3

(All revisions were carried out using track change)

The manuscript reports the investigation of the cannabinoid affinity of the neolignans extracts from Magnolia grandiflora. While appreciating the technical soundness of the study and the clarity of the results, the reviewer has some doubts regarding the novelty of the reported results since the presence of these neolignans into the extracts from Magnolia species was already reported and their activity on the cannabinoid receptors was already document along with the study on the affinity of these biphenol derivatives on other therapeutically relevant targets such as GABAA and GPR55 receptors, kinases and other enzymes. Probably the only original part is the molecular docking but the reviewer is not sure that it can render the paper publishable.  

Response: We appreciate the reviewer's comment and agree that it may seem little unclear for the general readers, so we have now revised our Introduction, Discussion and Conclusion sections to highlight the novelty of the work as well as the potential future direction. Plese see our responses to the comments from reviewer 1, above.

Concerning the modelling studies, they are technically correct and the results are described in detail. The reviewer would suggest to start the docking analyses with a redocking study of the bound ligands to assess if the utilized computational approach is able to reproduce the experimental structure.

Response: We have included the validation protocol of our docking procedure, used to check the quality of the approach, in the main text of the revised manuscript. The docking protocol was validated by self-docking (re-docking) in which the native ligands, 8D3 and E3R, were docked into their corresponding protein structures, CB1 and CB2, respectively. Further, we calculated the RMSD between docked poses and experimental poses of the native ligands with the CB1 and CB2 receptors, respectively. The overlay of experimental poses of the native ligands with the docked poses showed an identical conformation with very small RMSD differences of 0.35 Å and 0.64 Å, respectively.

Finally, the reviewer thinks that the manuscript can be considered only if the Authors convincingly explain which is the novelty of their study compared to the already published data.

Response: We have updated the manuscript to better include the novelty of the work as well as the potential future directions in the revised manuscript. The main novelty and future directions of this study have been highlighted in the revised manuscript as follows: “An in-depth understanding of the SAR of the isolated compounds as well as the additive effect between compounds 2 and 3 has been reported for the first time, supported via the in vitro data and the in silico docking studies that take advantage of the first CB2 X-ray crystal structure which, was released in 2020.”

We also included the following sentence in the conclusion section to make a more clear and concise summary of our current work. “Methylation of 2 at the p-position and the replacement of the hydroxyl group with a methoxy substituent increase the displacement activity towards CB1 as shown in Table 1. Similarly, an equal (1:1) mixture of 2 and 3 showed additive displacement activity towards the tested receptors compared to either 2 or 3 alone, which in turn provides an explanation for the strong displacement activity of the n-hexane extract”.

Round 2

Reviewer 1 Report

The whole manuscript should be checked for typos and grammatical errors, 

Reviewer 3 Report

The manuscript was revised and now its novelty is better understandable,

The manuscript can now deserve publication